# Seasonal patterns of ecological uniqueness of anuran metacommunities along different ecoregions in Western Brazil

**Karoline Ceron**[1]*, **Diego J. Santana**[1,2], **Francisco Valente-Neto**[1]

**1** Programa de Pós-Graduação em Ecologia e Conservação, Instituto de Biociências, Universidade Federal de Mato Grosso do Sul, Cidade Universitária, Campo Grande, Mato Grosso do Sul, Brazil, **2** Instituto de Biociências, Universidade Federal de Mato Grosso do Sul, Cidade Universitária, Campo Grande, Mato Grosso do Sul, Brazil

* adenomera@gmail.com

**Data Availability Statement:** All relevant data are within the manuscript and its Supporting Information files.

## Abstract

Beta diversity can be portioned into local contributions to beta diversity (LCBD), which represents the degree of community composition uniqueness of a site compared to regionally sampled sites. LCBD can fluctuate among seasons and ecoregions according to site characteristics, species dispersal abilities, and biotic interactions. In this context, we examined anuran seasonal patterns of LCBD in different ecoregions of Western Brazil, and assessed their correlation with species richness and if environmental (climatic variables, pond area and ecoregions) and/or spatial predictors (spatial configuration of sampling sites captured by distance-based Moran's Eigenvector Maps) would drive patterns of LCBD. We sampled anurans in 19 ponds in different ecoregions in the Mato Grosso do Sul state, Western Brazil, during one dry and one rainy season. We found that LCBD patterns were similar between seasons with sites tending to contribute in the same way for community composition uniqueness during the dry and rainy season. Among studied ecoregions, Cerrado showed higher LCBD values in both seasons. In addition, LCBD was negatively correlated with species richness in the dry season. We also found that LCBD variation was explained by ecoregion in the dry season, but in the rainy season both environmental and spatial global models were non-significant. Our results reinforce the compositional uniqueness of the Cerrado ecoregion when compared to the other ecoregions in both seasons, which may be caused by the presence of species with different requirements that tolerate different conditions caused by seasonality.

## Introduction

Understanding the organization of species diversity through space and time is one of the main scopes of community ecology [1]. Species diversity can be divided into gamma (regional diversity), alpha (local diversity), and beta components [2]. The latter (beta diversity) is the variation in species composition among sites within a region, first described by Whittaker [2, 3].

**Funding:** KC is grateful to Coordenação de Aperfeiçoamento de Pessoal de Nível Superior - Brasil (CAPES) - Finance Code 001 for her scholarship. DJS thanks CNPq (Conselho Nacional de Desenvolvimento Científico e Tecnológico) for his research fellowship (311492/2017-7). FVN was supported by grant number 88882.317337/2019-01, Coordenação de Aperfeiçoamento de Pessoal de Nível Superior (CAPES). This study was financed in part by the Coordenação de Aperfeiçoamento de Pessoal de Nível Superior - Brasil (CAPES) - Finance Code 001.

**Competing interests:** The authors have declared that no competing interests exist.

Such variation can be related to ecological processes, so analyzing beta diversity patterns can shed light on the comprehension of mechanisms underlying biodiversity patterns [4]. Beta diversity can be measured in different ways, including additive and multiplicative indices, dissimilarity measures, and beta diversity as variation in community structure among sampling units [4]. These methods include the partition of the variance of community data into species contributions to beta diversity (SCBD) and into local contributions to beta diversity (LCBD) [5]. LCBDs represent the degree of community composition uniqueness of a site compared to regionally sampled sites [5] and constitute an important tool to detect more unique sites in terms of community composition that can be used to guide conservation strategies and to detect keystone communities [6–8]. Keystone community is defined as communities with a disproportional positive impact relative to their weight in the metacommunity. One simple way to detect keystone communities is through the correlation between LCBD (a measure of the relative site impact in the metacommunity) and species richness (a measure of weight or size of local communities) [7–9]. Keystone communities would be those communities with high impact on metacommunity (high value of LCBD) and low value of species richness [8].

Local contributions to beta diversity can also be used to test if selection and/or dispersal-related processes explain biodiversity patterns [10–12]. Selection by both site characteristics and biotic interactions filters species from the regional species pool to occur in local communities. For example, in a study performed in Brazilian Atlantic Forest, Almeida-Gomes [13] found that larger forest patch sizes are important for amphibian persistence in fragmented landscapes. Dispersal also affects local community dynamics [9, 14]. High dispersal can reduce beta diversity among sites, homogenizing the metacommunity [9]. In contrast, low dispersion or dispersal limitation may increase beta diversity, because organisms cannot reach suitable sites and may increase the role of drift [15], as observed in the Brazilian Atlantic Forest [16, 17].

An increasing number of studies used the partitioning of beta diversity into LCBD and SCBD in a variety of plant and animal taxa to better understand biodiversity patterns [12, 18–22]. However, this method is still poorly explored among ecoregions, which are large units of land containing a distinct assemblage of natural communities and species [23, 24]. Typically, a given ecoregion is similar in structure along its extent, but shares few species with other ecoregions due to biogeographic barriers, species turnover caused by geographical distance, or by environmental and biotic selection [25, 26]. On a global scale, the relationship between dissimilarity in species composition and productivity varied according to ecoregion [27], but information on a finer scale is still scarce. The dissimilarity in species composition in a given region composed of different ecoregions can vary according to climate, vegetation type, disturbance regimes (e.g., fires), and migrations [23].

Besides the spatial variation in community composition, beta diversity can fluctuate over time in the same site, known as temporal beta diversity [28]. Understanding the temporal dynamics of communities can solve fundamental ecological processes, including effects of individual life histories on ecosystem change, the relative importance of biotic and abiotic factors in determining community structure, or how taxa and the networks in which they are embedded respond to environmental change [29]. Community composition changes through time occur due to gains and losses of species, as well as changes in species abundance, resulting from different ecological processes, including environmental seasonality [28, 30]. As consequence, LCDB value also fluctuate among seasons and its association with environmental and spatial factors can change among periods [31]. For example, Tolonen [31] found that drivers of compositional uniqueness of aquatic macroinvertebrates change between spring and autumn, which was mainly related to species life cycle events. The explained variation of compositional uniqueness by environmental variables (e.g., pH, particle size and stream width) decreased from spring to autumn, while the explained variation by the spatial variables increased notably [31]. Similarly,

Kong [32] shown that compositional uniqueness of fish changes between the dry and rainy seasons because of the presence of particular species moving back and forth from floodplain habitats. Thus, seasonal variation in compositional uniqueness depend on the life history of organism model and physical characteristics of the study area.

Understanding compositional uniqueness variation between seasons and its drivers may help to identify sites and species with high conservation values or sites that need to be restored [5]. Indeed, assessing variation in composition uniqueness among sites and seasons can improve our understanding on processes that generate and maintain biodiversity. The midwestern Brazil location has a highly seasonal variation in environmental conditions in the Atlantic Forest, Cerrado, Chaco and, Pantanal ecoregions. This region allows us to explore seasonal patterns of compositional uniqueness and compare the relative importance of the potential mechanisms explaining those patterns.

Neotropical anurans are considered excellent ecological models because they are locally abundant and their sampling is relatively easy [33]. Anurans are particularly susceptible to environmental and spatial factors because they have permeable skin, a biphasic life cycle, unshelled eggs and limited dispersal [34]. Most of them are dependent on ponds or water bodies for tadpoles development and adults reproduction. Considering that anuran biodiversity is highly threatened, suffering a severe global decline by virtue of diseases, climate change, and habitat loss [17, 35, 36], understanding spatial and temporal patterns may be highly useful for biodiversity conservation and for detecting sites that disproportionally contribute to regional species pool relative to species richness [5, 7, 8].

We examined anuran seasonal patterns (dry and rainy seasons) of compositional uniqueness (LCBD) in different ecoregions of Western Brazil and their correlation with species richness, thus elucidating possible keystone communities. We also assessed if environmental (climatic variables, pond area and ecoregions) and/or spatial predictors (spatial configuration of sampling sites captured by distance-based Moran's Eigenvector Maps) would drive patterns of LCBD. We expected that LCBD would differ among ecoregions for the dry season, and no difference would be found in LCBD for the rainy season. This expectation is based on the low water availability in dry season compared to the rainy season, when all ecoregions tended to be equal in terms of water availability. This water restriction in the dry season would filter species in naturally seasonally dry ecoregions, such as the Cerrado and Chaco [37], where water availability is a constraint for many species in the dry season [37], leading to more unique communities. We also expected that this filter would be more intensive in the Cerrado because this ecoregion is not close to floodplains that may maintain water availability during the dry season. The Chaco region is close to the Pantanal and both occupy the area under influence of Paraguay Basin flood pulses, which would provide water to anuran reproduction throughout the year. In this way, we expected that the Cerrado ecoregion would have higher values of LCBD compared to other ecoregions in the dry season. We also hypothesized that LCBD variation would be driven by environmental variables in the dry and rainy seasons, but the total amount of variation would be higher in the dry season.

## Material and methods

### Study area

We sampled anurans in 19 ponds located in Mato Grosso do Sul state, covering the Atlantic Forest, Chaco, Cerrado, and Pantanal ecoregions in Brazil (*sensu* Olson [23], Fig 1 and S1 Table). Typically, the dry season ranges from April to September, and the rainy season extends from October to March in the region. The Atlantic Forest and Cerrado ecoregions support the highest species richness and rates of endemism, and they have been undergoing huge forest

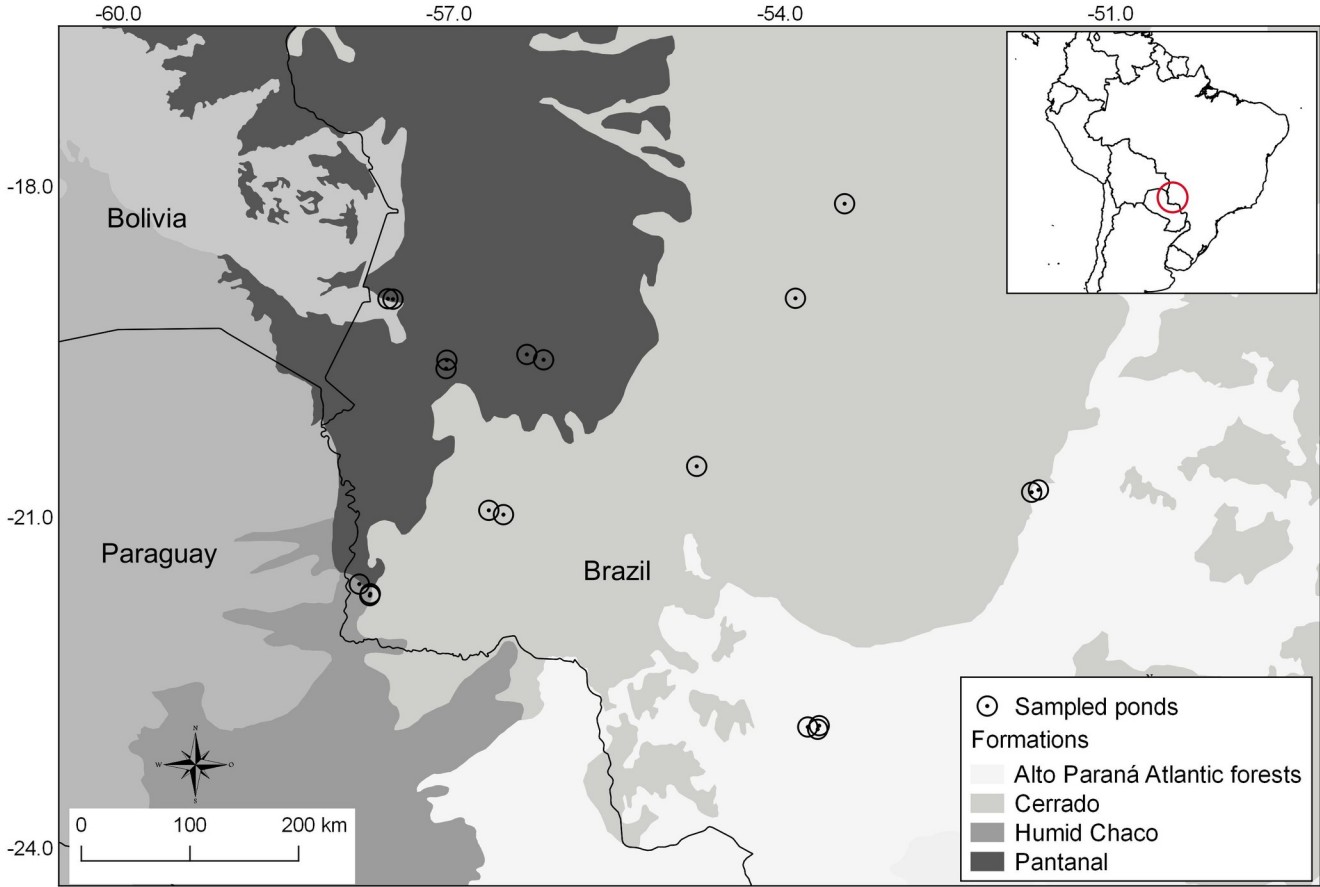

**Fig 1. Location of the sampled ponds in West Brazil for each ecoregion (Atlantic Forest, Chaco, Cerrado, and Pantanal).**

loss, being classified as hotspots of biodiversity [38, 39]. Atlantic Forest is characterized by heterogeneous and highly diverse plant species, with lowland, montane, semideciduous, and deciduous forests, but most of them are represented by small fragments [39, 40]. Semideciduous parts of the domain shared many species with neighbouring ecoregions (e.g., Cerrado) [41] and receive in the study region around 1313 mm/year of rainfall [42].

The Cerrado ecoregion is characterized by an extremely variable physiognomy, ranging from open grassland to forest with a discontinuous grass layer [43]. The overall amount of rainfall in the study region of the Cerrado is 1,424 mm/year [42]. The Chaco ecoregion is one of the most threatened subtropical woodland savannas in the world [44, 45]. Vegetation comprises xerophytic forests, alternating with patches of secondary woodlands and scrubs, and in temporarily flooded areas; the vegetation is typically composed of sclerophyllous grasslands. The Chaco ecoregion receive in the study region around 1,161 mm per year of rainfall [42]. The Cerrado and Chaco ecoregions are considered seasonally dry tropical forest, meaning that rainfall is less than c. 1800mm per year, with a period of at least 5–6 months receiving less than 100mm [37]. Pantanal is one of the largest wetlands in the world and is comprised of major vegetation formations: flood-free ridges (ancient levees) inhabited by trees, seasonally flooded plains with grasslands, and water bodies with aquatic macrophytes [46]. Although species diversity is not particularly high and endemism is practically absent, the region is notable for its abundance of wildlife [47]. Annual rainfall in the studied area of the Pantanal is around 1,177 mm [42]. Among the sampled sites, Cerrado is the only one that did not exhibit flood

pulses during the rainy season. Cerrado and Pantanal ecoregion show the higher values of precipitation seasonality (55.54 and 59.01 coefficient of variation, respectively) in relation to Atlantic Forest (46.64 coefficient of variation) and Chaco (45.45 coefficient of variation) [42]."

We sampled three ponds in Chaco (CH), five each in Cerrado (CE) and Atlantic Forest (semideciduous forest) (AF), and six in Pantanal (PA), during 2017 and 2018 (Fig 2 and S1 Table). Each pond constituted a replicate. The minimum distance among ponds was 500 m between CE3 and CE4. The remaining ponds were far more than 1 km distance from each

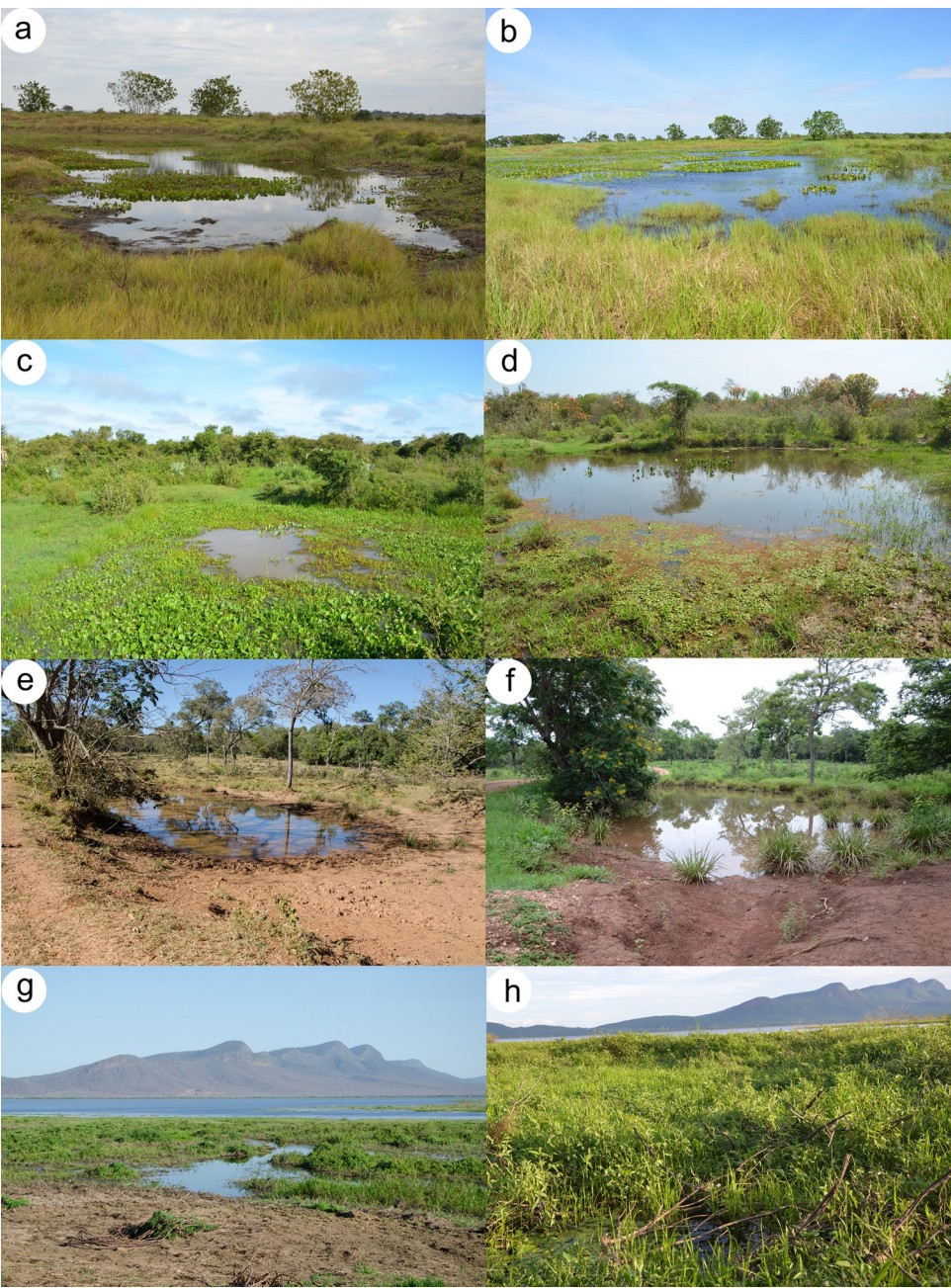

**Fig 2.** Some sampled sites during the dry and rainy season respectively in a–b) Atlantic Forest, c–d) Chaco, e–f) Cerrado, and g–h) Pantanal.

other. Each area was surveyed for one day per season during one dry and one rainy season, totalizing six hours of sampled effort per pond per season. We sampled anurans by active search [48] and visual and acoustic encounters conducted during time limited transects [49]. Samplings started on sunset and extended through midnight.

## Ethics statement

Anuran sampling was conducted under the permission of Brazilian wildlife regulatory service (SISBIO # 56729–1). The specimen manipulation was carried out following the recommendations of CEUA-UFMS protocol (# 838/2017). The collected individuals were sacrificed with the application of 5% lidocaine on the skin and fixed in 10% formalin, with later conservation in 70% alcohol.

## Environmental predictors

We used the location of each pond to extract 19 climatic variables from the BioClim database [42]. These variables cover different aspects of the mean and seasonal variability of temperature and precipitation (for more details see S2 Table). Climate predictors were extracted from raster files with 30 arc-second resolution using 'raster' package [50] in R version 3.5.0 [51]. For each location, we averaged each climatic variable over a 2000 m buffer zone to reduce the effect of uncertainty in study location. In addition, we chose this radius because the home range size of anurans can reach up to 2000 m [52].

Climatic variables were summarized by local contribution to environmental heterogeneity (LCEH), method developed by Castro [53]. To estimate LCEH for each site, we used standardized Euclidean distance [54]. Similar to LCBD, sites with high LCEH have singular environmental conditions while sites with low values have common environmental conditions. In addition to LCEH, we also included three dummy variables representing ecoregion specificities other than climatic (e.g., vegetational structure) and pond area as environmental predictors.

## Spatial predictors

We used distance-based Moran's eigenvector maps (dbMEM) on sampling sites' latitude and longitude [55, 56]. First, the minimum spanning tree distance that keeps all sites connected was calculated and used as a truncation threshold to construct the truncated matrix. This matrix was submitted to a Principal Coordinate Analysis (PCoA), and we selected the eigenvectors with significant patterns of spatial autocorrelation, i.e., with significant ($P < 0.05$) and positive Moran's I [57]. The eigenvectors represent spatial structures of relationships among the sampled sites, from broad to fine-scale patterns [57, 58]. We used the selected eigenvectors (MEMs) as spatial predictors in data analyses.

## Data analysis

We used the method described by Legendre and De Cáceres [5] to estimate both total beta diversity (BDtotal) and local contribution to beta diversity (LCBD). A community composition matrix (abundance data) was Hellinger transformed and then used to estimate BDtotal as the unbiased total sum of square of the species composition data. The BDtotal will assess LCBD, which is the relative contribution of each sampling unit to beta diversity, i.e., the division of sum of squares corresponding to each sampling unit by the total sum of squares. LCBD was calculated for dry (LCBDdry) and rainy (LCBDrainy) seasons independently.

We used Pearson correlation to assess if LCBD patterns of dry and rainy seasons were correlated. We also used Pearson correlation to assess the relationship between LCBD and species

richness. If a negative correlation between LCBD and richness is found, we may detect key-stone communities as those that have high LCBD (impact) and low richness (weight) [6, 8].

We used forward selection as implemented by Blanchet *et al.* [59] for significant global models. To select variables from an explanatory matrix, forward selection requires significance (p<0.05) and R2adj have to be below the global R2adj [59]. In this way, the explained variance is not overestimated, preventing the inflation of Type I error [59]. For non-significant global, we did not proceed with forward selection and variation partitioning, reporting just significant global model after forward selection. If both global models were significant, we used variation partitioning to divide the LCBD variation of each season into four components: pure environmental component [a], the amount of variation shared by environmental component and spatial component [b], pure specific spatial component [c] and non-explained variation (residual) [d]. The significance [a] and [c] were tested via permutation-based (1000 permutations) tests of partial multiple regressions models.

To perform all analyses, we used R language and the packages 'vegan' [60] 'packfor' [61] and 'adespatial' [62].

## Results

We sampled a total of 43 species and 1488 individuals distributed in Atlantic Forest (species = 20; individuals = 296), Cerrado (n = 25; 297), Chaco (n = 21; 289) and in Pantanal (n = 23; 606). On average, species richness tended to be higher in the Chaco ecoregion ($\bar{x}$= 10.3), followed by Atlantic Forest ($\bar{x}$= 8.8), Cerrado ($\bar{x}$= 7.4), and Pantanal ($\bar{x}$= 6.3) (S1 Table). *Dendropsophus nanus* was the most abundant species in Atlantic Forest (n = 64), Cerrado (n = 63), and in Pantanal (n = 138), and *Lysapsus limellum* was the most abundant species in Chaco (n = 62). Overall, species richness was higher during the wet season (n = 37) than the dry season (n = 32), as well as the total abundance (772 and 716, respectively). Atlantic Forest had 19 species in the wet season and 11 species in the dry season, while Cerrado had 21 and 14 species, in the wet season and dry seasons, respectively. Chaco had 18 and 14 species, and Pantanal 16 and 17 species, respectively for the rainy and the dry seasons. Of the sampled species, *Boana albopunctata*, *B. geographica*, *Leptodactylus furnarius*, *L. labyrinthicus*, *Phyllomedusa sauvagii*, *Pristimantis dundeei* were registered only in the Cerrado ecoregion, *Adenomera dyptix*, *L. latrans*, *L.* aff. *fuscus*, and *Scinax acuminatus* were registered only in the Pantanal, *Physalaemus biligonigerus*, *L. elenae*, *L. bufonius* and *Rhinella major* were registered only in the Chaco and *Dendropsophus sanborni*, *Elachistocleis bicolor* and *Scinax squalirostris* were registered only in the Atlantic Forest. The total beta diversity for the dry period was 0.60. The mean local contribution to beta diversity in this season was 0.052 (ranging from 0.024 to 0.097) (Fig 3A). Sites with the highest values (LCBD> = 0.080) had significant LCBDs (four sites, all in the Cerrado ecoregion), whereas sites with values lower than 0.080 had non-significant LCBDs. Cerrado sites had higher LCBD values than sites in other ecoregions. LCBD was negatively correlated with species richness in the dry season (Pearson correlation = -0.46, p = 0.04) (S2 Fig). In the rainy period, the total beta diversity was slightly lower compared to the dry season (BD total = 0.55). The mean local contribution to beta diversity in the rainy period was 0.052 (ranging from 0.030 to 0.100) (Fig 3B). Sites with the highest values in this period (LCBD> = 0.080) had significant LCBDs (two sites, one in the Cerrado and the other in the Pantanal ecoregions), whereas sites with values lower than 0.080 had no significant LCBDs. The pattern of higher LCBD in Cerrado sites was maintained in the rainy season (Fig 3). Contrary to the dry period, the relationship between LCBD and richness was not significantly correlated in the rainy season (Pearson correlation = 0.09, p = 0.69) (S2 Fig). LCBD values from dry period were significantly correlated with rainy season (Pearson correlation = 0.56, p = 0.01) (S1 Fig), demonstrating that similar sites contribute in the same way to compositional uniqueness (Fig 3).

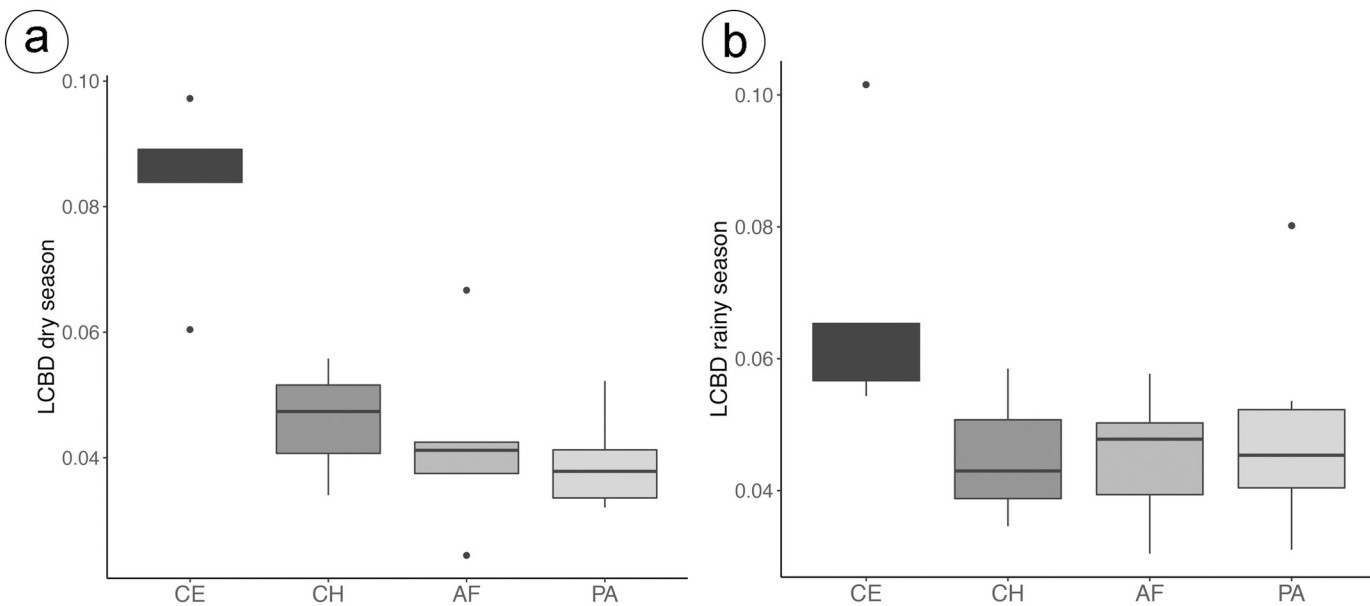

**Fig 3. Local contributions to beta diversity (LCBD) values for the dry and rainy seasons from the four ecoregions sampled (AF = Atlantic Forest, CH = Chaco, CE = Cerrado, and PA = Pantanal).**

The environmental global model was significant for the dry period, (p = 0.001) and the Cerrado ecoregion was the variable selected. Distance-based Moran's eigenvector maps generated three eigenvectors, all of them with positive and significant spatial correlation. Spatial global model was also significant (p = 0.008) and MEM3 was selected to be included in the variation partitioning. Pure environmental component composed by Cerrado ecoregion [a] significantly explained variance in LCBD values (p = 0.002; adjusted $R^2$ = 0.29), whereas pure spatial component composed by MEM3 [c] was not significant to explain LCBD variation in the four ecoregions (p = 0.20; adjusted $R^2$ = 0.01). The shared component between environmental and spatial components explained 42% of variation in LCBD values and the unexplained variation in LCBD values corresponded to 27%. In the rainy season, both environmental and spatial global models were not significant (environmental: F = 2.15, p = 0.22; spatial: F = 2.37, p = 0.11), and, consequently, we did not proceed with variation partitioning (Table 1).

## Discussion

In this study we found that LCBD patterns were similar between seasons, i.e., sites tended to contribute in the same way for community composition uniqueness during the dry and rainy season, contrary to our hypothesis. In addition, LCBD was negatively correlated with species

**Table 1.** Results of the partial redundancy analysis of site uniqueness for anurans during the dry season, where [a] pure environmental component, [b] the amount of variation shared by environmental component and spatial component, [c] pure specific spatial component and [d] non-explained variation (residual).

| | | | | [a] | [b] | | [c] | [d] |
|---|---|---|---|---|---|---|---|---|
| | **Env selected** | **Spa selected** | **R2adj** | **F** | **R2adj** | **R2adj** | **F** | **R2adj** |
| LCBD Dry | Dummy_Cerrado | MEM3 | **0.29** | **19.33**[**] | 0.42 | 0.01 | 1.80 | 0.27 |

[a] The explained variation for component b was -0.21 and for this reason the residual presented in the table is 0.50. According to Legendre & Legendre (2012) negative explained variance should be interpreted as 0.00. [*] 0.05<p<0.01; [*] 0.01<p<0.001.

Bold: represents significant fractions. Results for the rainy season were omitted because both environmental and spatial global models were non-significant.

richness in the dry season. Among studied ecoregions, Cerrado showed higher LCBD values in both seasons, despite lower values during the rainy season. We also found that LCBD variation was explained by pure environmental variables (ecoregion) in the dry season, but models were non-significant during the rainy season.

For both seasons, local contributions to beta diversity were higher in Cerrado sites than in Atlantic Forest, Chaco and Pantanal, partially confirming our hypothesis. Cerrado is considered one of the world's 'hotspots' for biodiversity conservation because of its high endemism and its high rates of habitat conversion and biodiversity loss [38]. In relation to anurans, Cerrado has high species richness and endemism with assemblages from different lineages, which is likely a result of its contact with four South American ecoregions: Amazonia, Atlantic Forest, Caatinga, and Chaco [63, 64].

On the other hand, the similarity of LCBD values among Atlantic Forest, Chaco, and Pantanal might be related to their similarity in floodplain areas and by the elevated number of common and well-distributed species, such as *L. limellum*. In the study area, these ecoregions are strongly influenced by great rivers such as the Paraná and Paraguay, which flood seasonally and can act as migration routes for modern floras and faunas [65]. Moreover, sites of Atlantic Forest, Chaco, and Pantanal in this study can be considered transition zones because they are located at the boundaries between biogeographic regions and represent areas of biotic overlap, which are promoted by historical and ecological changes that allow the mixture of different biotic elements [66, 67]. Thus, each area could allow the entrance of well-distributed species coming from the surrounding ecoregions, in turn affecting the distribution of species and LCBD values in the core of the study sites.

We detected that sites tended to contribute in the same way to beta diversity in both seasons. Sampled sites in Atlantic Forest, Chaco, and Pantanal are composed of ponds that are more connected to adjacent ponds in the rainy season and isolated during the dry season. Conversely, in dry season Cerrado ponds experience the decreasing the amount of water available in ponds, forcing anurans to aestivate or seek shelter [68], and favour species that do not depend on water or are more adapted to desiccation (e.g., *P. dundeei* and *L. furnarius*) [69, 70], increasing LCBD values. In the rainy season, the greater water availability in Cerrado sites tends to decrease the difference between LCBD values from those values of other ecoregions. As a result, the seasonal LCBD patterns in the Cerrado ecoregion between seasons may be driven by drought periods and species requirements. Considering all these patterns, Cerrado sites may be keystone areas because of their disproportional contribution to regional species pool relative to their species richness in the dry season [7, 8].

Environmental heterogeneity is an important driver in metacommunity theory, with organisms tracking environmental variation over the region via dispersal [71]. In our study, sites tended to contribute in the same way for community composition uniqueness during the dry and rainy season, but the factors explaining each seasonal pattern differed. These results indicate that understanding the mechanisms responsible for beta diversity patterns is distant from to be cleared, as more unique habitats and marked seasons are not necessarily the ones harbouring more unique communities [53]. The different requirements among species can lead to some differences in community responses to environmental variables, when dispersal is limited or restrained by seasons [72]. In the dry season, our results indicated that LCBD variation was related to pure environmental variables (ecoregion characteristics) and by shared component (spatially structured environmental variables). The effect of environmental filters is stronger during the dry than the rainy season, filtering species that tolerate water restrictions [73]. Anurans can minimize energy use during dry periods and may aestivate or hibernate once the availability of resources and reproductive habitats decrease due to lower humidity or temperatures [74]. Also, species that require less water (e.g., viviparous species that do not depend on

water for reproduction, *P. dundeei*) tend to appear in the dry season, mainly in Cerrado, increasing LCBD values in this ecoregion. Similar results were obtained for anurans from Amazonian sites, where the compositional uniqueness was more strongly associated with the environment [21], and for macrophytes in China when diversity patterns were driven mainly by spatially structured environmental determinism [75]. Therefore, pronounced seasonal environments may impose a fluctuating selection on life history traits, selecting species according to their requirements in the dry season due to desiccation.

During the rainy season, optimal conditions are experienced by the majority of anurans and environmental selection is less pronounced. The elevated rainfall triggers breeding in the majority of anurans [76], many of them widely distributed and habitat generalists, like *Dendropsophus nanus* and *D. minutus*. Anuran communities are more similar in this season, leading to similar LCBD values. For example, ponds in Pantanal and Chaco are more connected to adjacent sites in the rainy season, where flood pulses are more pronounced [77]. Flood pulses are also an important force for semideciduous areas of Atlantic forest near the Paraná River, promoting dispersal and the homogenization of communities. These pulses tend to connect ponds, favoring species dispersal among sites within each ecoregion (Pantanal, Chaco, and Atlantic Forest) [78, 79] and potentially between some of them, such as Pantanal and Chaco. This connection provides large areas available for breeding, which minimizes resource competition among individuals, favoring dispersion of species. These factors may be related to the non-significance of environmental and spatial models during this season. Besides to provide large areas for breeding the rainy season also provide a great amount of prey to anurans [80], because the composition of invertebrates in an environment change throughout a year in relation to climatic variations, different requirements among species, and life history stages [81]. Thus, the non-significance of environmental and spatial models can be related to the optimal conditions of species during this season, with species not being constrained by environmental or spatial filters.

Combining site-specific contributions to beta diversity in different seasons, we identified sites that consistently harbored unique communities, contributing to the maintenance of a regional species pool. Based on our analyses, Cerrado sites can be considered as keystone communities, because they have a disproportional contribution to the regional species pool in the dry season. The presence of a unique set of species composition, derived from its high endemism relative to the other ecoregions, increases the local contribution to beta diversity of Cerrado. Despite its enormous importance for species conservation and the provision of ecosystem services, only 19.8% of the native vegetation of Cerrado remains undisturbed [43]. The change in land uses as livestock and pastures is the main driver to deforestation of this hotspot and will drive ~480 endemic plant species to extinction [43, 82]. Thus, this elevated exploitation may reduce biodiversity in Cerrado sites, and consequently, would cause great effects in the anuran metacommunity. Therefore, to maintain the role of Cerrado as keystone areas, we suggest the identification and mapping more of these sites in order to preserve the regional biodiversity. In addition, through environmental education, owners of these areas should be made aware of the importance of these areas for regional diversity and should help maintain the ecological process associated with these species.

## Supporting information

**S1 Fig. Pearson correlation between LCBD values during dry and rainy seasons.** Sites abbreviation can be seen in the S1 Table.
(JPG)

**S2 Fig.** Pearson correlation between richness and LCBD values during dry (a) and rainy seasons (b). To studied sites abbreviation see S1 Table.
(JPG)

**S1 Table. Ponds sampled during the years of 2017 and 2018 in West Brazil.**
(DOCX)

**S2 Table. Raw climatic variables.** All temperature and precipitation values were extracted from BioClim (http://worldclim.org/current) for each studied community. All values were averaged over the surrounding 2km to help buffer uncertainty in the reported locations. Variables indicates the name of the climatic variable in the respective date source.
(DOCX)

# Acknowledgments

We thank the owners of Fazenda Barranco Alto, APA Baía Negra, Fazenda Patolá, Fazenda 7 Irmãos, Parque Estadual das Nascentes do Rio Taquari, Parque Estadual das Várzeas do Rio Ivinhema, RPPN Brejo Bonito, and Base de estudos do Pantanal UFMS for allowing access to their properties. KC is grateful to Liliana Piatti, Franscisco Severo-Neto, Andressa Figueiredo, and all Mapinguari lab members for their essential help during fieldworks. The authors are very grateful to Fritz Hertel for his constructive criticism and English review.

# Author Contributions

**Conceptualization:** Karoline Ceron, Diego J. Santana, Francisco Valente-Neto.

**Data curation:** Karoline Ceron.

**Formal analysis:** Francisco Valente-Neto.

**Investigation:** Karoline Ceron.

**Supervision:** Diego J. Santana, Francisco Valente-Neto.

**Writing – original draft:** Karoline Ceron.

**Writing – review & editing:** Karoline Ceron, Diego J. Santana, Francisco Valente-Neto.

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
