## [Decision Letter · Decision Letter 0]

14 Aug 2020

PONE-D-20-20490

Seasonal patterns of ecological uniqueness of anuran metacommunities along different Brazilian ecoregions

PLOS ONE

Dear Dr. Ceron,

I finally received the revision of two reviewers regarding your submission entitled "Seasonal patterns of ecological uniqueness of anuran metacommunities along different Brazilian ecoregions." (PONE-D-20-20490). The reviewers provided strong opposite suggestions about your manuscript, advising from rejection to acceptance. I tend to agree with one of the reviewers who highlighted several problems throughout the text and with the sampling design.

Nevertheless, and considering the relevance of the study for the subject, I propose you to make a major revision on your manuscript if you feel you can fix the problems raised by the reviewer. Specifically, you should demonstrate that the data are not spatially correlated, i.e., the sample sites are independent. Besides, the unbalanced number of samples between ecorregions should be taken into account in the analysis. Another critical point is the need to restructure the Abstract and Introduction, as indicated by the reviewer.

If you do not provide strong arguments to the reviewer's points, I cannot accept your paper in its current structure. When resubmitting your manuscript, please pay attention to all issues indicated by the reviewer and provide suitable rebuttals to any of his/her comments.

We look forward to receiving your revised manuscript.

Kind regards,

Ricardo Bomfim Machado, D.Sc.

Academic Editor

PLOS ONE

3. We note that Figure 1 in your submission contain map images which may be copyrighted. All PLOS content is published under the Creative Commons Attribution License (CC BY 4.0), which means that the manuscript, images, and Supporting Information files will be freely available online, and any third party is permitted to access, download, copy, distribute, and use these materials in any way, even commercially, with proper attribution. For these reasons, we cannot publish previously copyrighted maps or satellite images created using proprietary data, such as Google software (Google Maps, Street View, and Earth). For more information, see our copyright guidelines: http://journals.plos.org/plosone/s/licenses-and-copyright.

3.1.    You may seek permission from the original copyright holder of Figure 1 to publish the content specifically under the CC BY 4.0 license. 

3.2.    If you are unable to obtain permission from the original copyright holder to publish these figures under the CC BY 4.0 license or if the copyright holder’s requirements are incompatible with the CC BY 4.0 license, please either i) remove the figure or ii) supply a replacement figure that complies with the CC BY 4.0 license. Please check copyright information on all replacement figures and update the figure caption with source information. If applicable, please specify in the figure caption text when a figure is similar but not identical to the original image and is therefore for illustrative purposes only.

Reviewers' comments:

Reviewer's Responses to Questions

**Comments to the Author**

1. Is the manuscript technically sound, and do the data support the conclusions?

Reviewer #1: Yes

Reviewer #2: No

2. Has the statistical analysis been performed appropriately and rigorously? 

Reviewer #1: Yes

Reviewer #2: No

3. Have the authors made all data underlying the findings in their manuscript fully available?

Reviewer #1: Yes

Reviewer #2: No

4. Is the manuscript presented in an intelligible fashion and written in standard English?

Reviewer #1: Yes

Reviewer #2: Yes

5. Review Comments to the Author

Reviewer #1: In the manuscript entitled " Seasonal patterns of ecological uniqueness of anuran metacommunities along diferente Brazilian ecoregions”, the authors realized field research to verify anuran seasonal patterns of local contributions to beta diversity (LCBD) in different ecoregions of West Brazil, its correlation with species richness and assessed if environmental and/or spatial predictors would drive patterns of LCBD.

The authors sampled anurans in 19 ponds located in Mato Grosso do Sul state, covering the Atlantic Forest (5 ponds), Chaco (3 ponds), Cerrado (5 ponds), and Pantanal ecoregions in Brazil (6 ponds). The methodology used is adequate. The statistical tests used are also adequate. The results are presented clearly, with tables and figures adequate and sufficient to understand the text.

The results indicate the uniqueness of the Cerrado anurofauna when compared to the other two ecoregions, which can be caused by the occurrence of species with different thresholds to tolerate seasonality.

So, I consider the manuscript suitable to be published in Plos One.

Reviewer #2: Assessment of the paper entitled “Seasonal patterns of ecological uniqueness of anuran metacommunities along different Brazilian ecoregions” for Plos One (PONE-D-20-20490)

Main comments

In this study, the authors aimed to evaluate anuran seasonal patterns (dry and rainy seasons) of compositional uniqueness (LCBD, local contributions to beta diversity) in different ecoregions of West Brazil (Atlantic Forest, Chaco, Cerrado, and Pantanal) and correlation of LCBD and species richness. They also assessed if environmental predictors and/or spatial predictors drive patterns of LCBD across these ecoregions.

Since its introduction in 2013 (Legendre & De Cáceres, 2013), ecological uniqueness (or LCBD) has gained scientific attention due to its potential to highlight sites that harbor singular species compositions (either based on presence or abundance data). This information can be used to better inform decision-makers on more suitable areas to put efforts of conservation or management. However, there is a need to better understand the correlation and drivers of ecological uniqueness and environmental and spatial variables across a range of taxa and ecosystems across the globe. This study has its merit for providing information on this regard. However, the study has some limitations that need to be better addressed before publication. I present these issues as follows.

Abstract

I think the abstract could be restructured since some methods are presented before the main objective. Some detailed information could be added as well. Besides, there is no result on the correlation LCBD-species richness, which was the main objective. I strongly suggest that authors use the 300 words available to improve the description of the study presented in the abstract.

Introduction

There is no good argumentation on seasonal patterns to lead the reader to the authors’ objective and hypothesis. This is a main issue the authors need to address by adding some examples of the relationship between diversity patterns or LCBD and seasons, especially for the region studied. There is a lot of information on ecoregions but not on seasonality effects or patterns. However, the introduction lacks good detailing on the Brazilian ecoregion sampled. The authors state that “mid-western Brazil allows us to explore compositional uniqueness and its drivers in four ecoregions” but do not detail them in the introduction. Therefore, the reader cannot understand the authors’ assumptions or main ideas. Besides, the organisms used as study models are not presented in detail as well. Anurans are organisms dependent on ponds or water bodies, but authors fail to present the context of their ideas for this specific fauna. As a whole, the introduction has not a good deal with both the study area and the organism’s model of study. In addition, there is no argumentation to lead the reader to understand the aim to correlate LCBD and species richness. These issues should be better presented and contextualized aiming a clear and objective understanding by readers.

Sampling design

The authors sampled anurans in a total of 19 ponds: 3 ponds in Chaco, 5 in Cerrado, 5 in Atlantic Forest (semideciduous forest), and 6 in Pantanal, during 2017 (dry season) and 2018 (rainy season). However, based on Figure 1, a total of 7 ponds are located in transition zones between ecoregions, which is not a good approach. For instance, the 3 ponds of Chaco are only in the transition zone with Pantanal. In fact, the authors samples only a limited range of each ecoregion when they did. The sample design is also unbalanced in this has serious implications to beta diversity estimations, such as LCBD. I bring these issues because later the authors state that climatic variables were average due to “the effect of uncertainty in study location”. This averaging method based on 2 km buffer also makes some ponds 500 m apart to be autocorrelated. As authors state, ponds up to 2 km apart are not independent because the “home range size of anurans can reach up to 2000 m”. Each pond was surveyed for one day per season (active search and visual and acoustic encounters) during one dry and one rainy season, totalizing six hours of sampled effort per pond per season. This is a very limited sampling camping to fully collect representative samples. These issues rise some serious concerns. For instance, the authors did not evaluate or present any estimate of sample completeness. This is required due to limited sampling and unbalanced sample design. These major issues have severe impacts on diversity patterns and hence on study results.

Minor issues

Line 21-22. A general definition of LCBD would be nice here, before presenting some expected drivers of LCBD.

L 22. Change to “according to”

L 24. Well, Atlantic Forest is the most eastern ecoregion or vegetation domain in Brazil – we can see by its name. So, why do authors state that Atlantic Forest is in West Brazil? I am not sure about that. What about Parana Forest and Araucaria angustifolia Forests? See Morrone (2006) and associated works by this author.

L 27. Why does correlate only to species richness? There is plenty of information on this correlation (mostly negative), so why not include other metrics like abundance- or biomass-related metrics, beta diversity components (e.g. replacement, richness difference, nestedness), functional diversity metrics? Besides, the authors are too generic in mentioning ‘environmental’ and ‘spatial’ predictors. Could you provide some examples? Which environmental and spatial variables were evaluated?

References

Legendre, P. & De Cáceres, M. (2013) Beta diversity as the variance of community data: dissimilarity coefficients and partitioning. Ecology Letters, 16, 951-963.

Morrone, J.J. (2006) Biogeographic areas and transition zones of Latin America and the Caribbean islands based on panbiogeographic and cladistic analyses of the entomofauna. Annual Review of Entomology, 51, 467-494.

6. PLOS authors have the option to publish the peer review history of their article (what does this mean?). If published, this will include your full peer review and any attached files.

Reviewer #1: **Yes: **Rogério Bastos

Reviewer #2: No

---

## [Author Response · Author response to Decision Letter 0]

21 Aug 2020

Rebuttal letters referring to ms PONE-D-20-20490

Dear Dr. Ricardo Bomfim Machado

Academic Editor of PLoS ONE Journal

Thank you for the invitation to submit a revised version of our manuscript. We appreciate the constructive criticisms and suggestions by the editor and referees. We considered all points raised by the editor and reviewers, and carefully addressed them as described below. 

In summary, we explained that the predictors used captured different structures of spatial autocorrelation, tested for spatial autocorrelation in the residuals of our final model for the dry season and demonstrated that the unbalanced number of sampling sites did not affect our findings. We also would like to highlight that the environmental variable selected for the dry season was Cerrado ecoregion and not Pantanal ecoregion, as stated in the previous version. We corrected this issue in this revised version. We also improved the summary and introduction as recommended. 

We believe that this new version of the manuscript incorporates all main aspects pointed out by the reviewers, and hope that it meets the high standards for publication in Plos One.

Editor

1. Nevertheless, and considering the relevance of the study for the subject, I propose you to make a major revision on your manuscript if you feel you can fix the problems raised by the reviewer. Specifically, you should demonstrate that the data are not spatially correlated, i.e., the sample sites are independent. Besides, the unbalanced number of samples between ecorregions should be taken into account in the analysis. Another critical point is the need to restructure the Abstract and Introduction, as indicated by the reviewer.

R: We are grateful for these suggestions. You are right about the spatial correlation. However, we were interested in spatial autocorrelation in our data and because of that we modelled distance-based Moran’s eigenvector maps (dbMEM) on sampling sites’ latitude and longitude. Spatial autocorrelation in community composition may be caused by dispersal (our interest here) and random processes, as well as unmeasured environmental variables (Leibold & Chase 2018). The analysis resulted in three eigenvectors, all of them with positive and significant spatial correlation (Moran’s I) (Figure 1 in pdf. version). Note that the three MEMs captured a range of spatial autocorrelation that were included in our analysis and did not affect our response variable (LCBD) for both seasons, reinforcing that spatial autocorrelation did not explain LCBD variation. To emphasize that our data is not affected by spatial autocorrelation, we ran a Moran’s I test for residuals of the dry season final model (LCBD dry season~Cerrado ecoregion) and the test was not significant (observed Moran’s I=-0.06; expected Moran’s I=-0.05; p=0.75). In this way, this final model meets the independence assumption of linear models and the residuals are not affected by spatial autocorrelation. For the rainy season, we did not find any predictor to be significant, and we did not run Moran’s I test. Thus, considering our results of no relationship between LCBD for both seasons with MEM’s and the absence of spatial autocorrelation in the residuals of the final model for the dry season, we did not see any problem in our data regarding spatial autocorrelation. 

Figure 1. Eingenvectors resulted from Distance-based Moran’s eigenvector maps (dbMEM) with positive and significant spatial autocorrelation. Blue and red depict positive and negative scores, respectively, and the size of dots represents the magnitude of scores.

Thank you for the comment regarding unbalanced number of samples between ecoregions. This would be a real problem if the variance is not equal among ecoregions, violating another assumption of linear regression (homogeneity of variances) and requiring to model a variance structure, such as generalized least square model does. To verify this issue, we checked homogeneity of variance of LCBD values between ecoregions using Levene’s test. For both seasons, we found that our data meet the homogeneity of variance assumption (dry season: F value=0.308; p=0.82; rainy season=0.30, p=0.82). We also emphasized that in figure 3 in the manuscript (also presented below as Figure 2 in this response letter - pdf. version) visually showed that the variability in LCBD among ecoregions is nearly the same. 

Figure 2. Local contributions to beta diversity (LCBD) values for the dry and rainy seasons from the four ecoregions sampled (AF=Atlantic Forest, CH=Chaco, E=Cerrado, and PA=Pantanal).

In addition, we correlated the mean LCBD of each ecoregion with the respectively sample size. Despite we recognize that this analysis uses a low number of samples for each season (number of ecoregions=4), results indicated no correlation between mean LCBD and sample size (dry season: Pearson correlation=0.003, p=0.99; rainy season: Pearson correlation=0.25, p=0.74) (Figure 3). Finally, we restructured the introduction and the abstract following the suggestions. 

Figure 3. Relationship between mean LCBD and number of samples of each ecoregion for dry and rainy seasons.

Reviewer #2

1. Abstract. I think the abstract could be restructured since some methods are presented before the main objective. Some detailed information could be added as well. Besides, there is no result on the correlation LCBD-species richness, which was the main objective. I strongly suggest that authors use the 300 words available to improve the description of the study presented in the abstract

R: We improved the abstract as recommended. Now it reads (lines 21-39): “Beta diversity can be portioned into local contributions to beta diversity (LCBD), which represents the degree of community composition uniqueness of a site compared to regionally sampled sites. LCBD can fluctuate among seasons and ecoregions according to site characteristics, species dispersal abilities, and biotic interactions. In this context, we examined anuran seasonal patterns of LCBD in different ecoregions of Western Brazil and assessed its correlation with species richness and if environmental (climatic variables, pond area and ecoregions) and/or spatial predictors (spatial configuration of sampling sites captured by distance-based Moran’s Eigenvector Maps) would drive patterns of LCBD. We sampled anurans in 19 ponds in different ecoregions in the Mato Grosso do Sul state, Western Brazil, during one dry and one rainy season. We found that LCBD patterns were similar between seasons with sites tending to contribute in the same way for community composition uniqueness during the dry and rainy season. Among studied ecoregions, Cerrado showed higher LCBD values in both seasons. In addition, LCBD was negatively correlated with species richness in the dry season. We also found that LCBD variation was explained by ecoregion in the dry season, but in the rainy season both environmental and spatial global models were non-significant. Our results reinforce the compositional uniqueness of the Cerrado ecoregion when compared to the other ecoregions in both seasons, which may be caused by the presence of species with different requirements that tolerate different conditions caused by seasonality.”. 

2. Introduction. There is no good argumentation on seasonal patterns to lead the reader to the authors’ objective and hypothesis. This is a main issue the authors need to address by adding some examples of the relationship between diversity patterns or LCBD and seasons, especially for the region studied. There is a lot of information on ecoregions but not on seasonality effects or patterns. 

R: We agree with the reviewer and we added a paragraph addressing the effects of temporal changes on LCBD. Now it reads (lines 85-104): “Besides the spatial variation in community composition, beta diversity can fluctuate over time in the same site, known as temporal beta diversity [28]. Understanding the temporal dynamics of communities can illuminate fundamental ecological processes, including effects of individual life histories on ecosystem change, the relative importance of biotic and abiotic factors in determining community structure, or how taxa and the networks in which they are embedded respond to environmental change [29]. Community composition changes through time occur due to gains and losses of species, as well as changes in species abundance, resulting from different ecological processes, including environmental seasonality [28,30]. As consequence, LCDB value also fluctuate among seasons and its association with environmental and spatial factors can change among periods [31]. For example, Tolonen [31] found that drivers of compositional uniqueness of aquatic macroinvertebrates change between spring and autumn, which was mainly related to species life cycle events. The explained variation of compositional uniqueness by environmental variables (e.g., pH, particle size and stream width) decreased from spring to autumn, while the explained variation by the spatial variables increased notably [31]. Similarly, Kong [32] shown that compositional uniqueness of fish changes between the dry and rainy seasons because of the presence of particular species moving back and forth from floodplain habitats. Thus, seasonal variation in compositional uniqueness depend on the life history of organism model and physical characteristics of the study area.”

3. Introduction. However, the introduction lacks good detailing on the Brazilian ecoregion sampled. The authors state that “mid-western Brazil allows us to explore compositional uniqueness and its drivers in four ecoregions” but do not detail them in the introduction. 

R: We are grateful for this comment. We added a sentence in the introduction explaining why ecoregions are interesting to study patterns of compositional uniqueness (lines 105-112): “Understanding compositional uniqueness variation between seasons and its drivers may help researchers to identify sites and species with high conservation values or sites that need to be restored [5]. Indeed, assessing variation in composition uniqueness among sites and seasons may help to improve our understanding of the processes that generate and maintain biodiversity. The mid-western Brazil location has a highly seasonal variation in environmental conditions in the Atlantic Forest, Cerrado, Chaco and, Pantanal ecoregions. This region allows us to explore seasonal patterns of compositional uniqueness and compare the relative importance of the potential mechanisms explaining those patterns.” Also, in M&M section, there is a detailed description of ecoregions characteristics in lines 142-166.

4. Introduction. Besides, the organisms used as study models are not presented in detail as well. Anurans are organisms dependent on ponds or water bodies, but authors fail to present the context of their ideas for this specific fauna. As a whole, the introduction has not a good deal with both the study area and the organism’s model of study.

R: We appreciate this comment. We added a paragraph explaining the organisms of study. Now it reads (lines 113-122): “Neotropical anurans are considered excellent ecological models because they are locally abundant and their sampling is relatively easy [33]. Anurans are particularly susceptible to environmental and spatial factors because they have as permeable skin, a biphasic life cycle, unshelled eggs and limited dispersal [34]. Most of them are dependent on ponds or water bodies for tadpoles development and adults reproduction. Considering that anuran biodiversity is highly threatened, suffering a severe global decline by virtue of diseases, climate change, and habitat loss [17, 35, 36], understanding spatial and temporal patterns may be highly useful for biodiversity conservation and for detecting sites that disproportionally contribute to regional species pool relative to species richness [5, 7, 8].”

5. Introduction. In addition, there is no argumentation to lead the reader to understand the aim to correlate LCBD and species richness. These issues should be better presented and contextualized aiming a clear and objective understanding by readers.

R: We added a sentence to turn it cleaner in the end of introduction first. Now it reads (lines 56-61): “Keystone community is defined as communities with a disproportional positive impact relative to their weight in the metacommunity. One simple way to detect keystone communities is through the correlation between LCBD (a measure of the relative site impact in the metacommunity) and species richness (a measure of weight or size of local communities) [6–8]. Keystone communities would be those communities with high impact on metacommunity (high value of LCBD) and low value of species richness” We also better explain our aim related to this issue (lines 122-124): “We examined anuran seasonal patterns (dry and rainy seasons) of compositional uniqueness (LCBD) in different ecoregions of West Brazil and its correlation with species richness, thus elucidating possible keystone communities”

6. Sampling design. The authors sampled anurans in a total of 19 ponds: 3 ponds in Chaco, 5 in Cerrado, 5 in Atlantic Forest (semideciduous forest), and 6 in Pantanal, during 2017 (dry season) and 2018 (rainy season). However, based on Figure 1, a total of 7 ponds are located in transition zones between ecoregions, which is not a good approach. For instance, the 3 ponds of Chaco are only in the transition zone with Pantanal. In fact, the authors samples only a limited range of each ecoregion when they did. The sample design is also unbalanced in this has serious implications to beta diversity estimations, such as LCBD. I bring these issues because later the authors state that climatic variables were average due to “the effect of uncertainty in study location”. This averaging method based on 2 km buffer also makes some ponds 500 m apart to be autocorrelated. As authors state, ponds up to 2 km apart are not independent because the “home range size of anurans can reach up to 2000 m”. 

R: We are grateful for your comment. Unfortunately, our sampling design is restricted by the accessibility of fieldwork areas and by the availability of preserved ecoregion areas in the Mato Grosso do Sul state. As a floodplain, Pantanal accessibility during the rainy season is extremely difficult and expensive, so we had to select areas with easy access for both dry and rainy season. As Cerrado is full of crops and, consequently, roads, the movement in this ecoregion is easier, allowing us to better distribute the samples. The Chaco ecoregion in Brazil comprehends a small piece of land in Mato Grosso do Sul state, naturally bordered by the Pantanal. Because it is a small area, the Chaco ecoregion is practically a transition zone with Pantanal. The Chaco area covers the municipalities of Caracol, Bela Vista e Porto Murtinho and the ecoregion is dominated by farms with few areas available to research activities. In the same way, the portion of Atlantic Forest of Mato Grosso do Sul is restricted to the region of Dourados and Três Lagoas municipalities, regions where agrobusiness is well developed, which turn it hard to find a preserved area suitable for our research. 

Thank you for the comment regarding unbalanced number of samples between ecoregions. As we respond to the editor, this would be a real problem if the variance is not equal among ecoregions, violating another assumption of linear regression (homogeneity of variances) and requiring to model a variance structure, such as generalized least square model does. To verify this issue, we checked homogeneity of variance of LCBD values between ecoregions using Levene’s test. For both seasons, we found that our data meet the homogeneity of variance assumption (dry season: F value=0.308; p=0.82; rainy season=0.30, p=0.82). We also emphasized that manuscript figure 3 (also presented below as Figure 2 in this response letter - in pdf. version) visually showed that the variability in LCBD among ecoregions is nearly the same. 

We also correlated the mean LCBD of each ecoregion with the respectively sample size. Despite we recognize that this analysis uses a low number of samples (number of ecoregions=4) for each season, results indicated no correlation between mean LCBD and sample size (dry season: Pearson correlation=0.003, p=0.99; rainy season: Pearson correlation=0.25, p=0.74) (Figure 3 above in this response letter - in pdf. version).

We emphasize that the use of unbalanced sampled design to assess LCBD patterns is a common approach in the literature. For example, Vilmi et al. (2017) (Diversity and Distributions, 23:1042–1053) ended up with 492 stream and 290 lake sites with information on diatom community composition, and compared these data in relation to LCBD. Biguezoton et al. (2016) (Parasites & Vectors, 9:43) compared LCBD among four areas in Benin and Burkina Faso, with different number of sampling sites in each area. See also Datry et al. (2016) (Freshwater Biology 61, 1335–1349), and Lopes et al. (2014) (PLoS ONE 9(10):e109581).

LCBD dry season(mean) LCBD rainy season(mean) Sites

AF 0.042 0.045 5

CE 0.083 0.068 5

PA 0.039 0.049 6

CH 0.045 0.045 3

We also would like to explain your criticism about spatial autocorrelation. First, we were indeed interested in spatial autocorrelation in our data and because of that we modelled distance-based Moran’s eigenvector maps (dbMEM) on sampling sites’ latitude and longitude. Spatial autocorrelation in community composition may be caused by dispersal (our interest here) and random processes, as well as unmeasured environmental variables (Leibold & Chase 2018). The analysis resulted in three eigenvectors, all of them with positive and significant spatial correlation (Moran’s I) (see Figure 1 above in this letter - in pdf. version). Note that the three MEMs captured a range of spatial autocorrelation that were included in our analysis and did not affect our response variable (LCBD) for both seasons, reinforcing that spatial autocorrelation did not explain LCBD variation. To emphasize that our data is not affected by spatial autocorrelation, we ran a Moran’s I test for residuals of the dry season final model (LCBD dry season~Cerrado ecoregion) and the test was not significant (observed Moran’s I=-0.06; expected Moran’s I=-0.05; p=0.75). In this way, this final model meets the independence assumption of linear models and the residuals are not affected by spatial autocorrelation. For the rainy season, we did not find any predictor to be significant, and we did not run Moran’s I test. Thus, considering our results of no relationship between LCBD for both seasons with MEM’s and the absence of spatial autocorrelation in the residuals of the final model for the dry season, we did not see any problem in our data regarding spatial autocorrelation. 

We also re-ran our statistical analysis to show that the inclusion of points separated by 500m did not affect our conclusion. To do so, we removed site CE4 from the analysis and the results were maintained. For the dry season, the results showed that environmental global model was significant (F=7.30, p=0.006) and Cerrado ecoregion was selected by forward selection procedure (F=46.64, p=0.001). Also, spatial global model was significant (F=9.00, p=0.007) and MEM3 was selected by forward selection procedure (F=16.22, p=0.003). Pure environmental component composed by Cerrado ecoregion [a] significantly explained variance in LCBD values (p=0.007; adjusted R2=0.22), whereas pure spatial component composed by MEM3 [c] was not significant to explain LCBD variation in the four ecoregions (p=0.18; adjusted R2=0.01). We also re-ran this analysis removing point CE5 and the conclusion was the same (only pure environmental component composed by Cerrado ecoregion explained LCBD variation). 

For the rainy season and removing CE4, analyses showed that both environmental (F=1.34, p=0.3) and spatial global models were not significant (F=2.25, p=0.13). Removing CE5, the results are the same (environmental global model F=2.30, p=0.1; spatial global models F=2.36, p=0.12).

7. Sampling design. Each pond was surveyed for one day per season (active search and visual and acoustic encounters) during one dry and one rainy season, totalizing six hours of sampled effort per pond per season. This is a very limited sampling camping to fully collect representative samples. These issues rise some serious concerns. For instance, the authors did not evaluate or present any estimate of sample completeness. This is required due to limited sampling and unbalanced sample design. These major issues have severe impacts on diversity patterns and hence on study results.

R: We added to Table S1 a column showing the sample coverage of all sites. On average, the sample coverage was 78.14%, indicating a good sampled effort (Figure 4 in pdf. version). 

Table S1. Ponds sampled during the years of 2017 and 2018 in West Brazil.

Name site Site Formation Lat Long Pond área (m²) Wet richness Dry richness Sample coverage

Brejo Bonito CE1 Cerrado -20.5377 -54.7548 6617 7 2 61.15

Camapuã CE2 Cerrado -19.0142 -53.8591 866 5 4 78.41

Mimosa 01 CE3 Cerrado -20.9659 -56.524 1355 12 5 100.00

Mimosa 02 CE4 Cerrado -20.9685 -56.5211 1018 7 5 73.09

Taquari CE5 Cerrado -18.1571 -53.413 5770 6 5 89.21

Chaco 01 CH1 Chaco -21.6929 -57.7169 1736 11 7 53.09

Chaco 02 CH2 Chaco -21.6065 -57.8163 802 9 8 94.17

Chaco 03 CH3 Chaco -21.71 -57.7209 1146 11 11 92.64

Três Lagoas 01 AF1 Atlantic Forest -20.7513 -51.6544 839 5 2 73.17

Três Lagoas 02 AF2 Atlantic Forest -20.7727 -51.7158 2888 4 4 83.57

Ivinhema 01 AF3 Atlantic Forest -22.9218 -53.6571 1809 12 6 60.48

Ivinhema 02 AF4 Atlantic Forest -22.9008 -53.7471 652 12 5 60.30

Ivinhema 03 AF5 Atlantic Forest -22.889 -53.6439 797 11 8 65.10

Barranco Alto 01 PA1 Pantanal -19.5724 -56.1548 2061 7 7 100.00

Barranco Alto 02 PA2 Pantanal -19.5719 -56.144 4250 4 8 100.00

BEP 01 PA3 Pantanal -19.5752 -57.0217 1116 6 5 53.25

BEP 02 PA4 Pantanal -19.5765 -57.0187 434 6 6 97.33

Baía Negra 01 PA5 Pantanal -19.0222 -57.5106 6670 7 9 74.47

Baía Negra 02 PA6 Pantanal -19.0184 -57.5564 4052 8 8 75.30

Figure 4: Sampling coverage of all sites studied in Mato Grosso do Sul state, Brazil

8. Line 21-22. A general definition of LCBD would be nice here, before presenting some expected drivers of LCBD.

R: We added a general definition of LCBD as recommended. Now it reads (lines 21-24) “Beta diversity can be portioned into local contributions to beta diversity (LCBD), which represents the degree of community composition uniqueness of a site compared to regionally sampled sites. LCBD can fluctuate among seasons and ecoregions according to site characteristics, species dispersal abilities, and biotic interactions”. 

9. L 22. Change to “according to”

R: Sentence modified as suggested. 

10. L 24. Well, Atlantic Forest is the most eastern ecoregion or vegetation domain in Brazil – we can see by its name. So, why do authors state that Atlantic Forest is in West Brazil? I am not sure about that. What about Parana Forest and Araucaria angustifolia Forests? See Morrone (2006) and associated works by this author.

R: Thank you for the comment, and you are right, the Atlantic Forest is the most eastern ecoregion or vegetation domain in Brazil. In our study region (Mato Grosso do Sul state), Atlantic Forest is located in the south-eastern region (as you can see in Fig. 1 in the manuscript). Lines 28-30 states that “We sampled anurans in 19 ponds in different ecoregions in Western Brazil (…)” but we believe that, to make this sentence clearer we can state as: “We sampled anurans in 19 ponds in different ecoregions in the Mato Grosso do Sul state, Western Brazil (…)”. We follow the regionalization proposed by Olson 2011, cited in line 139 of M&M section. He considers as ecoregions: Pantanal, Humid Chaco, Cerrado and Alto Paraná Atlantic Forest. 

162. L 27. Why does correlate only to species richness? There is plenty of information on this correlation (mostly negative), so why not include other metrics like abundance- or biomass-related metrics, beta diversity components (e.g. replacement, richness difference, nestedness), functional diversity metrics? 

R: We understand that we can use many biodiversity dimensions here. However, we only correlated LCBD with richness to elucidate possible keystone communities in the study region. Keystone community is defined as communities with a disproportional positive impact relative to their weight in the metacommunity, which can better inform decision-making in conservation (Mouquet et al. 2013). Valente-Neto et al. (2020) proposed this correlation as a simple way to estimate keystone communities, because LCBD is a measure of contribution of local sites to the metacommunity (a measure of the relative site impact in the metacommunity) and species richness is a measure of size of local communities (weight). Following Mouquet et al (2013) keystone communities would be those communities with high impact on metacommunity (high value of LCBD) and low value of species richness. We added in the first paragraph of the introduction the reasoning to include this correlation (Lines 56-61). As we stated in M&M (lines 218-220), if a negative correlation between LCBD and richness is found, we may detect keystone communities as those that have high LCBD (impact) and low richness (weight) [6–8]. 

11. Besides, the authors are too generic in mentioning ‘environmental’ and ‘spatial’ predictors. Could you provide some examples? Which environmental and spatial variables were evaluated?

R: We agreed with your comment and modify the sentence to (lines 24- 28):“(…) and if environmental (climatic variables, pond area and ecoregions) and/or spatial predictors (spatial configuration of sampling sites captured by distance-based Moran’s Eigenvector Maps) would drive patterns of LCBD”. We also include this explanation in the aims paragraphs (lines 124-127).

REFERENCES

Leibold, M. A. & J. M. Chase, 2018. Metacommunity Ecology. Princeton University Press, Princeton.

Mouquet N, Gravel D, Massol F, Calcagno V. Extending the concept of keystone species to communities and ecosystems. Ecol Lett. 2013;16: 1–8.

Valente-Neto F, da Silva FH, Covich AP, de Oliveira Roque F. Streams dry and ecological uniqueness rise: environmental selection drives aquatic insect patterns in a stream network prone to intermittence. Hydrobiologia. 2020;847: 617–628.

---

## [Decision Letter · Decision Letter 1]

26 Aug 2020

PONE-D-20-20490R1

Seasonal patterns of ecological uniqueness of anuran metacommunities along different Brazilian ecoregions

PLOS ONE

Dear Dr. Ceron,

Thank you for submitting your manuscript to PLOS ONE. I received a positive evaluation from the reviewer regarding your last responses. I am now setting the manuscript as under "minor revision," but yet, there are two minor points (see below) that still need to be clarified. The first one is a slight modification on the title to have it clearer and more focused (see a suggestion below). The other point is the need to work a little bit on the hypothesis descriptions.

We look forward to receiving your revised manuscript.

Kind regards,

Ricardo Bomfim Machado, D.Sc.

Academic Editor

PLOS ONE

Reviewers' comments:

Reviewer's Responses to Questions

**Comments to the Author**

1. If the authors have adequately addressed your comments raised in a previous round of review and you feel that this manuscript is now acceptable for publication, you may indicate that here to bypass the “Comments to the Author” section, enter your conflict of interest statement in the “Confidential to Editor” section, and submit your "Accept" recommendation.

Reviewer #2: (No Response)

2. Is the manuscript technically sound, and do the data support the conclusions?

Reviewer #2: Yes

3. Has the statistical analysis been performed appropriately and rigorously? 

Reviewer #2: Yes

4. Have the authors made all data underlying the findings in their manuscript fully available?

Reviewer #2: Yes

5. Is the manuscript presented in an intelligible fashion and written in standard English?

Reviewer #2: Yes

6. Review Comments to the Author

Reviewer #2: Assessment of the paper entitled “Seasonal patterns of ecological uniqueness of anuran metacommunities along different Brazilian ecoregions” for Plos One (PONE-D-20-20490_R1)

Main comments

After reviewing this second version of the study, considering the response letter provided by the authors, I had the opportunity to better understand the context and the choices made by the authors. Many issues have been clarified and I appreciate the authors’ efforts to provide appropriate answers to my comments. But two important issues need to be addressed properly, in my opinion.

Title. The title is not accurate enough because it leads the reader to a broader scope of ecoregion of Brazil as it is: “Seasonal patterns of ecological uniqueness of anuran metacommunities along different Brazilian ecoregions”, but the sampling of these in parts of these ecoregions was limited to Mato Grosso do Sul state. Brazil has more than 20 ecoregions distributed across 6 main vegetation domains. In this paper, only 4 were sampled and again I am not totally secure that those ponds on ecotones between two different ecoregions belong to one or another. The number of ecoregions sampled may be due to some disparities in map projections and sampling coordinates. Even if such problems are overcome, I would still suggest that the title be adjusted to: “Seasonal patterns of ecological uniqueness of anuran metacommunities along different ecoregions in Western Brazil” in order to describe the paper content more accurately.

Hypothesis/Prediction. The authors state that “LCBD patterns would differ between seasons due to the restriction of water availability in dry compared to rainy seasons. This would filter species in naturally dry ecoregions, such as Cerrado, where water availability is a constraint for many species [37], leading to more unique communities.” Later, the authors state that “The Chaco ecoregion receive around >700 mm per year of rainfall”, which is the lowest total amount of rain among ecoregions (Atlantic Forest: 1000 mm/y; Cerrado: 800-2000 mm/y; Pantanal: 1089 mm/y). Besides, Chaco comprises xerophytic vegetation. Therefore, this hypothesis/prediction is not supported by the description of ecoregions and I had a hard time understanding this hypothesis/prediction. I think the authors need to be clearer in what sense they assess/characterize dry conditions among ecoregions to support the hypothesis/prediction.

7. PLOS authors have the option to publish the peer review history of their article (what does this mean?). If published, this will include your full peer review and any attached files.

Reviewer #2: No

---

## [Author Response · Author response to Decision Letter 1]

1 Sep 2020

Dear Dr. Ricardo Bomfim Machado

Academic Editor of PLoS ONE Journal

Thank you for the invitation to submit a new revised version of our manuscript. We appreciate the constructive criticisms and suggestions by the editor and referee. We considered all points raised by the editor and reviewer, and carefully addressed them as described below. 

In summary, we modified our title based on the reviewer suggestion and we improved our hypothesis about the restriction of water availability involving Cerrado and Chaco ecoregions.

We believe that this new version of the manuscript incorporates all main aspects pointed out by the reviewers, and hope that it meets the high standards for publication in PLoS ONE.

Reviewer #2

1. Title. The title is not accurate enough because it leads the reader to a broader scope of ecoregion of Brazil as it is: “Seasonal patterns of ecological uniqueness of anuran metacommunities along different Brazilian ecoregions”, but the sampling of these in parts of these ecoregions was limited to Mato Grosso do Sul state. Brazil has more than 20 ecoregions distributed across 6 main vegetation domains. In this paper, only 4 were sampled and again I am not totally secure that those ponds on ecotones between two different ecoregions belong to one or another. The number of ecoregions sampled may be due to some disparities in map projections and sampling coordinates. Even if such problems are overcome, I would still suggest that the title be adjusted to: “Seasonal patterns of ecological uniqueness of anuran metacommunities along different ecoregions in Western Brazil” in order to describe the paper content more accurately.

R: We are grateful for this suggestion. We modified the title as recommended. Now it reads: “Seasonal patterns of ecological uniqueness of anuran metacommunities along different ecoregions in Western Brazil”. 

2. Hypothesis/Prediction. The authors state that “LCBD patterns would differ between seasons due to the restriction of water availability in dry compared to rainy seasons. This would filter species in naturally dry ecoregions, such as Cerrado, where water availability is a constraint for many species [37], leading to more unique communities.” Later, the authors state that “The Chaco ecoregion receive around >700 mm per year of rainfall”, which is the lowest total amount of rain among ecoregions (Atlantic Forest: 1000 mm/y; Cerrado: 800-2000 mm/y; Pantanal: 1089 mm/y). Besides, Chaco comprises xerophytic vegetation. Therefore, this hypothesis/prediction is not supported by the description of ecoregions and I had a hard time understanding this hypothesis/prediction. I think the authors need to be clearer in what sense they assess/characterize dry conditions among ecoregions to support the hypothesis/prediction.

R: We agree with your comment. First, these values of rainfall are the range for each ecoregion. For example, Cerrado is the second larger ecoregion in Brazil (and in South America) and is bordered by many ecoregions, including Atlantic Forest, Pantanal, Amazônia, and Caatinga, justifying this high range in rainfall. Also, both Cerrado and Chaco are considered seasonally dry tropical forest (Pennington et al. 2009), meaning that rainfall is less than c. 1800mm per year, with a period of at least 5-6 months receiving less than 100mm (Pennington et al. 2009). This seasonality of Cerrado and Chaco justify our hypothesis of higher LCBD values for Cerrado and Chaco (see the small change we made below). We updated the mean annual precipitation in the methods including values for the studied region (Fick & Hijmans 2017) and we also included the mean precipitation seasonality for each ecoregion (lines 154, 162, 166-169, 174). Despite Cerrado showed the highest and Chaco the least mean annual precipitation, the values between ecoregions are not so different. On the other hand, Cerrado is the second ecoregion in terms of seasonality and Chaco is the third. Although Pantanal had the highest seasonality, water is more available in this ecoregion due to large rivers (e.g. Rio Paraguai), and many natural lakes, known locally as baías. This would provide refuges for anuran species in the dry season, different from the Cerrado ecoregion. Our point here is that the ecoregions did not greatly vary in terms of rainfall, but they did in terms of seasonality, which may influence water availability for anurans.

We modified our hypothesis to (lines 127- 141): “We expected that LCBD would differ among ecoregions for the dry season, and no difference would be found in LCBD for the rainy season. This expectation is based on the low water availability in dry season compared to the rainy season, when all ecoregions tended to be equal in terms of water availability. This water restriction in the dry season would filter species in naturally seasonally dry ecoregions, such as the Cerrado and Chaco [37], where water availability is a constraint for many species in the dry season [37], leading to more unique communities. We also expected that this filter would be more intensive in the Cerrado because this ecoregion is not close to floodplains that may maintain water availability during the dry season. The Chaco region is close to the Pantanal and both occupy the area under influence of Paraguay Basin flood pulses, which would provide water to anuran reproduction throughout the year. In this way, we expected that the Cerrado ecoregion would have higher values of LCBD compared to other ecoregions in the dry season. We also hypothesized that LCBD variation would be driven by environmental variables in the dry and rainy seasons, but the total amount of variation would be higher in the dry season”.

We also added in methodology the following information (lines 167-169): “The Cerrado and Chaco ecoregions are considered seasonally dry tropical forest, meaning that rainfall is less than c. 1800mm per year, with a period of at least 5-6 months receiving less than 100mm [37].”

Lines 187-190: “Cerrado and Pantanal ecoregion show the higher values of precipitation seasonality (55.54 and 59.01 coefficient of variation, respectively) in relation to Atlantic Forest (46.64 coefficient of variation) and Chaco (45.45 coefficient of variation) [42].”

References

Fick SE, Hijmans RJ. WorldClim 2: new 1‐km spatial resolution climate surfaces for global land areas. Int J Climatol. 2017;37: 4302–4315.

Pennington RT, Lavin M, Oliveira-Filho A. Woody Plant Diversity, Evolution, and Ecology in the Tropics: Perspectives from Seasonally Dry Tropical Forests. Annu Rev Ecol Evol Syst. 2009;40: 437–457. doi:10.1146/annurev

---

## [Decision Letter · Decision Letter 2]

15 Sep 2020

Seasonal patterns of ecological uniqueness of anuran metacommunities along different ecoregions in Western Brazil

PONE-D-20-20490R2

Dear Dr. Ceron,

We’re pleased to inform you that your manuscript has been judged scientifically suitable for publication and will be formally accepted for publication once it meets all outstanding technical requirements.

Kind regards,

Ricardo Bomfim Machado, D.Sc.

Academic Editor

PLOS ONE

Additional Editor Comments (optional):

Reviewers' comments:

Reviewer's Responses to Questions

**Comments to the Author**

1. If the authors have adequately addressed your comments raised in a previous round of review and you feel that this manuscript is now acceptable for publication, you may indicate that here to bypass the “Comments to the Author” section, enter your conflict of interest statement in the “Confidential to Editor” section, and submit your "Accept" recommendation.

Reviewer #2: All comments have been addressed

2. Is the manuscript technically sound, and do the data support the conclusions?

Reviewer #2: Yes

3. Has the statistical analysis been performed appropriately and rigorously? 

Reviewer #2: Yes

4. Have the authors made all data underlying the findings in their manuscript fully available?

Reviewer #2: Yes

5. Is the manuscript presented in an intelligible fashion and written in standard English?

Reviewer #2: Yes

6. Review Comments to the Author

Reviewer #2: (No Response)

7. PLOS authors have the option to publish the peer review history of their article (what does this mean?). If published, this will include your full peer review and any attached files.

Reviewer #2: No

---

## [Editor Report · Acceptance letter]

16 Sep 2020

PONE-D-20-20490R2 

Seasonal patterns of ecological uniqueness of anuran metacommunities along different ecoregions in Western Brazil 

Dear Dr. Ceron:

I'm pleased to inform you that your manuscript has been deemed suitable for publication in PLOS ONE. Congratulations! Your manuscript is now with our production department. 

Kind regards, 

on behalf of

Dr. Ricardo Bomfim Machado 

Academic Editor

PLOS ONE